# Evaluation of the Antifungal Activity of the *Licania Rigida* Leaf Ethanolic Extract against Biofilms Formed by *Candida* Sp. Isolates in Acrylic Resin Discs

**DOI:** 10.3390/antibiotics8040250

**Published:** 2019-12-04

**Authors:** Maria Audilene de Freitas, Adryelle Idalina Silva Alves, Jacqueline Cosmo Andrade, Melyna Chaves Leite-Andrade, Antonia Thassya Lucas dos Santos, Tatiana Felix de Oliveira, Franz de Assis G. dos Santos, Maria Daniela Silva Buonafina, Henrique Douglas Melo Coutinho, Irwin Rose Alencar de Menezes, Maria Flaviana Bezerra Morais-Braga, Rejane Pereira Neves

**Affiliations:** 1Laboratory of Medical Mycology Sylvio Campos, Department of Mycology, Federal University of Pernambuco-UFPE, Recife, PE 50670-901, Brazil; audbiologa@hotmail.com (M.A.d.F.); adryellealvees@gmail.com (A.I.S.A.); melynaleite@gmail.com (M.C.L.-A.); tatifoliveira13@gmail.com (T.F.d.O.); franz.assis@gmail.com (F.d.A.G.d.S.); danielabuonafina@hotmail.com (M.D.S.B.); rejadel@yahoo.com.br (R.P.N.); 2Laboratory of Microbiology and Molecular Biology, Department of Biological Chemistry, Regional University of Cariri—URCA, Crato, CE 63105-000, Brazil; andradejacquelinec@gmail.com (J.C.A.); hdmcoutinho@gmail.com (H.D.M.C.); 3Laboratory of Mycology applied of Cariri, Department of biological Sciences, Regional University of Cariri—URCA, Crato, CE 63105-000, Brazil; thassyalucas@hotmail.com (A.T.L.d.S.); flavianamoraisb@yahoo.com.br (M.F.B.M.-B.); 4Laboratory of Pharmacology and Molecular Chemistry, Department of chemical biology, Regional University of Cariri—URCA, Crato, CE 63105-000, Brazil

**Keywords:** biofilm, *Candida* sp., *Licania rigida*, acrylic resin disc

## Abstract

*Candida* sp. treatment has become a challenge due to the formation of biofilms which favor resistance to conventional antifungals, making the search for new compounds necessary. The objective of this study was to identify the composition of the *Licania rigida* Benth. leaf ethanolic extract and to verify its antifungal activity against *Candida* sp. and its biofilms. The composition identification was performed using the ultra-high performance liquid chromatography-quadrupole time-of-flight mass spectrometry (UPLC-QTOF-MS/MS) technique. The antifungal activity of extract and fluconazole against planktonic cells and biofilms was verified through the minimum inhibitory concentration (MIC) following biofilm induction and quantification in acrylic resin discs by reducing tetrazolic salt, with all isolates forming biofilms within 48 h. Six constituents were identified in the extract, and the compounds identified are derivatives from phenolic compounds such as flavonoids (epi) gallocatechin Dimer, epigallocatechin and gallocatechin, Myricetin-O-hexoside, Myricitrin, and Quercetin-O-rhamnoside. The extract reduced biofilm formation in some of the strains analyzed, namely *C. tropicalis* URM5732, *C. krusei* INCQS40042, and *C. krusei* URM6352. This reduction was also observed in the treatment with fluconazole with some of the analyzed strains. The extract showed significant antifungal and anti-biofilm activities with some of the strains tested.

## 1. Introduction

Yeasts from the *Candida* spp. genus are part of the normal skin, mouth, gastrointestinal tract, and genitourinary tract microbiota. However, these commensal microorganisms can become pathogenic if changes in host defense mechanisms occur [1,2]. *C. albicans* is the most commonly found strain in human infections among *Candida* species [3]. Other species can be infectious, but are found less frequently [2].

The treatment of these infections has become a challenge due to the eukaryotic nature of fungal cells, which are similar to their host cells, and the occurrence of factors conferring resistance to conventional antifungals. This is especially challenging in immunocompromised individuals [4,5].

An important virulence factor of *Candida* species is their recognized ability to form biofilms on biotic and abiotic surfaces [6]. These microbial communities are formed on surfaces and are embedded in an extracellular matrix which can be found adhered to living tissue or to the surface of different materials, such as acrylic resin prostheses [7]. Biofilms represent a reduction in the susceptibility of microorganisms to the action of most antimicrobial agents, contributing to the permanence of the infection [8].

Resistance to commercially available antifungals has increased in recent decades [9]. The search for new bioactive substances that are more effective and less toxic to users, or those which present a new mechanism of action, have become essential today. Natural products can be considered promising in the discovery of new antifungal drugs given their chemical diversity and bioactive properties [10].

*Licania rigida* Benth, popularly known as “oiticica”, belongs to the Chrisobalanaceae family and is distributed in tropical and subtropical regions [11]. However, Sothers [12] proposed a new classification, with this species being classified into a new genus, *Microdesmia*, and renamed *Microdesmia rigida* Benth. In popular medicine this species is used for its anti-inflammatory properties, although few studies validating its pharmacological potential exist. Moreover, its antimicrobial activity has also been reported, however, there are few studies demonstrating its action against fungal pathogens [13].

Given the above, the present study aimed to chemically characterize the *Licania rigida* leaf ethanolic extract using the ultra-high performance liquid chromatography-quadrupole time-of-flight mass spectrometry - UPLC-MS-ESI-QTOF technique, to evaluate its antifungal activity, as well as its effect on the treatment of biofilms formed by *Candida* species.

## 2. Results

Characterization of the components present in the *Licania rigida* leaf ethanol extract (EEFLr) was performed using the UPLC-ESI-QTOF-MS method in the negative ionic mode. Figure 1 shows the High-Performance Mass Spectrometry Chromatogram (UPLC-MS) of the *Licania rigida* leaf ethanolic extract where the chromatographic peaks from the compounds present in the extract are observed. Identification based on molecular mass, retention time, fragmentation pattern and literature data resulted in six of the 12 compounds shown in Table 1. The compounds identified are derivatives from phenolic compounds such as flavonoids (epi) gallocatechin Dimer, epigallocatechin and gallocatechin, Myricetin-O-hexoside, Myricitrin, and Quercetin-O-rhamnoside.

The evaluation of the EEFLr antifungal activity against the used strains was determined by the MIC (2048 µg/mL), which showed significant results against *C. krusei* URM5712, *C. albicans* ATCC90028, *C. krusei* URM4263, *C. krusei* URM6352, and *C. krusei* URM5840 species obtaining MIC values of 256 µg/mL, 256 µg/mL, 32 µg/mL, 64 µg/mL, and 32 µg/mL, respectively. The remaining *C. albicans* URM5900, *C. tropicalis* URM5732, *C. albicans* INCQS40006, and *C. krusei* INCQS40042 isolates presented MIC values ≥ 1024 µg/mL. Fluconazole was used as a control drug and its concentrations ranged from 1 to 64 µg/mL (Table 2).

All yeast isolates were able to form biofilms within 48 h in the biofilm induction assay, however, these varied in intensities. In the biofilm treatment assays, *C. tropicalis* URM5732 and *C. krusei* INCQS40042 isolates showed a reduction in biofilm formation when compared with both the control and fluconazole treatment (Figure 2).

The *C. albicans* ATCC90028 isolate did not present significant differences between the extract and fluconazole treatments. As for *C. krusei* URM6352, biofilm reduction was observed with both treatments, while a greater biofilm reduction was observed for *C. krusei* URM4263 and *C. krusei* URM5840 yeasts treated with fluconazole than those treated with the extract (Figure 2).

## 3. Discussion

The constituents present in EEFLr such as (epi) gallocatechin Dimer, epigalocatechin and Gallocatechi, Myricetin-O-hexoside, Myricitrin, Quercetin-O-rhamnoside were found by other techniques in previous studies whereas the epicatechin and quercetin compounds were found by HPLC-DAD analysis of the *Licania rigida* ethanolic extract in a study by Parra [16]. Moreover, Braca et al. [17] performed this same analysis with different species from the *Licania* genus, *L. apetalada*, *L. densiflora*, *L. heteromorfa*, *L. pittieri*, *L. pyrifolia* and *L. carii*, showing the presence of flavonoid derivatives such as quercetin and myricetin in their composition. The study by Soares Santos et al. [18] with the *Licania rigida* hydroalcoholic extract also found epicatechin in its composition, results which corroborate those found the present study.

In a study by Morais [19], the *Licania rigida* extract was found to possess anti-*Candida* activity, a result also observed in this study. Some mechanisms of action of flavonoids have been described, among which are the dysregulation of nucleic acid synthesis and the ability to cause mitochondrial dysfunction [20].

The results suggest that the antifungal activity observed in this study may be related to Flavonoids, compounds that were found in EEFLr, because these compounds are capable of forming complexes with soluble proteins that are present in the fungal cell walls [21].The lipophilic nature of flavonoids is also capable of disrupting fungal membranes [21,22] and may inhibit the budding process and decrease the Ca^+^ and H^+^ homeostasis [23]. Similarly, the antifungal action observed in the present study may have been due to the presence of these compounds in the extract.

*Candida* species present differences in terms of biofilm formation, resistance, morphological characteristics and extracellular matrix [24]. Mature biofilms are much more resistant to antifungal therapy and host immune factors compared to planktonic yeast cells [25]. This variability increases the challenge in finding an effective solution to address the threat from biofilms caused by these pathogens [24].

Fluconazole is the commonly used drug to treat *Candida* infections, where its action on biofilms has been reported [26]. In our study, there were significant differences in the treatment of biofilms formed by three strains treated with this antifungal. Although a reduction in biofilm biomass was observed with fluconazole treatment in the present study, resistance to this drug is increasing among *Candida* species [26,27].

Tobaldini-Valerio et al. [28] used the propolis extract against *Candida* species in both planktonic cells and biofilms, with the authors attributing the observed inhibitory activity to the flavonoids present in the extract. Phenolic compounds impair the growth and formation of *C. albicans* biofilms, possibly by suppressing genes responsible for adhesion and morphogenesis [29].

Biofilms formed by *C. albicans* URM 5900 and INQCS 40006 isolates showed variations in responses to standard antifungal treatment (Fluconazole). This may be due to the characteristics of isolates that are not shared by the species. Moreover, a *C. albicans* biofilm biomass reduction has been demonstrated in previous studies [26,30].

Other *Candida* non-*C*. *albicans* species, such as *C. krusei*, are intrinsically resistant to the antifungal Fluconazole [31]. Thus, it is observable that the MIC of these strains was lower or similar to the fluconazole. In the present study, a reduction in biofilms formed by the *C. krusei* URM5840 and URM4263 isolates after 48 h of exposure to fluconazole was observed. Although these strains are commonly resistant to this drug, the precise mechanism by which this occurs has not been completely elucidated [30].

The *C. tropicalis* biofilm exhibited a resistance to fluconazole, as observed in biofilms from different *Candida* species [32,33]. In the study by Rajasekharan et al. [34], a reduction in *C. tropicalis* biofilm was observed when was treated with the flavonoid quercetin, with this reduction also being observed in this study. Wang et al. [35], observed the reduction in the biofilm formation has been studied and one of the most indicated mechanisms is the reduction of hydrophobicity of the cell surface, reducing the aggregative potential and the formation of the biofilm. Other possible mechanisms include the reduced activity of proteasomal enzymes by polyphenolic compounds, such as flavonoids. The inactivation of these enzymes reduced biofilm formation [36].

However, a biphasic behavior can be observed in the literature, with some *Candida* strains being stimulated to form biofilm, as observed in our study. This enhancement of the biofilm formation can be attributed to biochemical and biological factors to protect the *Candida* cells against environmental stressors, as phytocompounds with antimicrobial properties. Matsumoto et al. and Wang et al. [35,37] both reported the enhancement in biofilm aggregation by phytocompounds, demonstrating that the inhibition of *Candida* biofilms could be a strain-specific effect.

Studies with the *Licania rigida* species are still initial and many of the biological and pharmacological properties of its compounds have yet to be analyzed in order to understand and confirm their therapeutic indications. This study has shown that the *Licania rigida* extract was effective at reducing the MIC as well as biofilms of some of the strains used, presenting the extract as a possible source of bioactive substances with antifungal activity.

## 4. Materials and Methods

### 4.1. Botanical Material Collection and Identification

Collection was carried out in sitio Cuncas, located in the municipality of Barro-CE, Northeast Brazil, under geographic coordinates 07°05′26′′ south latitude and 38°43′17′′ longitude west of Greenwich, during the month of April 2017, at 9:00. Following collection, an exsiccate was prepared for identification, which was deposited in the Herbarium Dárdano de Andrade Lima (Herbário Dárdano de Andrade Lima) under identification number 13.742.

### 4.2. Extract Preparation

For the *Licania rigida* Benth. leaf ethanol extract (EEFLr) preparation, fresh leaves, cut to increase their surface area, were used. Afterwards, these were added to Ethanol PA, where 1g of leaves were used for each 1 mL of the solvent, utilizing cold maceration for extraction [38]. The final product was stored in a vessel protected from light and air for 72 h, after which this was filtered and concentrated in a rotary evaporator (Q-344B—Quimis—Brazil—40 rpm, 60 °C). The extract obtained was stored under refrigeration at 6 °C for assays.

### 4.3. Compound Identification by Ultra-High Performance Liquid Chromatography Coupled to a Quadrupole/Time-of-Flight System (UPLC-QTOF)

The identification of phenolic compounds present in the extract was performed in an Acquity^®^ UPLC system coupled to a Quadrupole/Time-of-Flight (QTOF) system (Waters Corporation, Milford, MA, USA), kindly provided by the Laboratory of Chemistry and Natural Products, Embrapa Tropical Agroindustry (Laboratório de Química e Produtos Naturais, Embrapa Agroindústria Tropical; Fortaleza, Ceara). Chromatographic runs were performed with a Waters Acquity UPLC BEH column (150 × 2.1 mm; 1.7 μm), 40 °C fixed temperature, mobile water phases with 0.1% formic acid (A) and acetonitrile with 0.1% formic acid (B), gradient ranging from 2% to 95% B (15 min), 0.4 mL/min flow rate, and 5 μL injection volume. The ESI mode was acquired in the 110–1180 Da range, with a 120 °C fixed source temperature, 350 °C desolvation temperature, 500 L/h desolvation gas flow, 0.5 V extraction cone, 2.6 kV capillary voltage. The ESI+ mode was acquired in the 110–1180 Da range, with a 120 °C fixed source temperature, 350 °C desolvation temperature, 500 L/h desolvation gas flow and 3.2 kV capillary voltage. Leucine enkephalin was used as a lock mass. MSE (high energy mass spectrometry) was the mode of acquisition used. The instrument was controlled by the Masslynx^®^ 4.1 software (Waters Corporation, Milford, MA, USA).

### 4.4. Strains Utilized

The strains used were obtained from the Federal University of Pernambuco Culture collection - URM Micoteca (Cultura da Universidade Federal de Pernambuco—Micoteca URM) as well as from the National Institute of Quality and Health—Oswaldo Cruz (Instituto Nacional de Qualidade e Saúde—Oswaldo Cruz) in Rio de Janeiro. The following 9 isolates were used: *Candida krusei* URM6352, *Candida krusei* URM4263, *Candida krusei* URM5840, *Candida albicans* URM5900, *Candida krusei* URM5712, *Candida albicans* ATCC90028, *Candida tropicalis* URM5732, *Candida krusei* INCQS40042 and *Candida albicans* INCQS40006.

### 4.5. Inoculum Preparation for the Sensitivity Test

A 24-h culture of the tested yeasts was performed on Sabouraud Dextrose Agar (SDA) prepared using an initial inoculum suspension in 5 mL of sterile saline (NaCl, 0.85% saline) where the density was adjusted accordingly to the 0.5 MacFarland scale with 90% transmittance determined by spectrophotometry, using a wavelength at 530 nm. This procedure provides a standard yeast concentration containing 1 × 10^6^ to 5 × 10^6^ cells per ml, followed by a 1:100 dilution and then a 1:20 dilution of the standard suspension with RPMI 1640 medium (Roswell Park Memorial Institute Medium) that’s is a growth medium used in cell culture, resulting in a concentration between 5.0 × 10^2^ and 2.5 × 10^3^, where these grew at a temperature of 37 °C.

### 4.6. Antifungal Sensitivity Test

The in vitro antifungal sensitivity test was performed according to the conditions described in the M27-A3 [39] and M60 [40] documents. The *C. albicans* ATCC 90028 isolate was used as the control. The antifungal agent Fluconazole (Pfizer), diluted in RPMI 1640, with concentrations ranging between 64 and 0.125 μg/mL was used. The extract was initially diluted in 1 mL of DMSO and further diluted in distilled water, with concentrations ranging from 2048 to 4 μg/mL. For the sensitivity tests, 96-well flat microdilution plates (TPP; Trasadingen, Switzerland) were used. The inoculum was added to the wells with the antifungal drug and the natural product, with the plates being incubated at 35 °C for 24 h to determine the minimum inhibitory concentration (MIC), which was used in the yeast biofilm treatment [39,40].

### 4.7. Acrylic Resin Disk Preparations

The discs were prepared using autopolymerisable acrylic resin associated with a self-polymerized acrylic liquid (VIPIFLASH). These were prepared in molds with 5 × 1 mm diameter. Following preparation, the disks were autoclaved.

### 4.8. Solution Preparations for the Biofilm Test

The extract matrix solution was prepared by diluting 0.05 g in 1 mL of dimethylsulfoxide (DMSO; Sigma-Aldrich, St. Louis, MO 63103, USA). The extract was prepared at a 2048 μg/mL concentration. Fluconazole (Pfizer) was used as the reference antifungal drug at the 64 μg/mL concentration.

### 4.9. Evaluation of the Biofilm Formation Capacity in Acrylic Resin Discs

The methodology described by Berridge et al. [41] and Krom et al. [42] was used with the following modification: the addition of acrylic resin discs. The tested yeasts were cultured in yeast extract peptone (YPD) at 37 °C overnight, then suspended in RPMI medium buffered with HEPES (4-(2-hydroxyethyl)-1-piperazineethanesulfonic acid) and adjusted to a concentration of 10^6^ cells/mL. Subsequently, 100 μL was added to wells in the 96-well polystyrene plates containing the acrylic resin discs which were maintained at 37 °C for 2 h at 75 rpm (adhesion phase). Thereafter, wells containing the discs were washed three times with phosphate buffer (PBS) to remove non-adherent cells, and these were then kept for 48 h at 37 °C for further evaluation.

Biofilm quantification was performed using the tetrazolium salt reduction assay, where 20 μL, in the 5 μL to 1 mL of PBS buffer ratio, sterilized by membrane filtration (Millipore, 0.22 μL pores) was added to each well of the microtiter plate, including control wells. The plates were incubated in the absence of light at 37 °C for 18 h. Afterwards, the dye was aspirated and 200 μL of isopropanol was added. The plates were allowed to rest for 15 min then 100 μL of the contents from each well were transferred to a new microtiter plate for reading in a microplate reader with a 570 nm wavelength.

### 4.10. Biofilm Treatment

The methodology described by Berridge et al. [41] and Krom et al. [42] was used with the following modification: the addition of acrylic resin discs. Biofilms were formed on resin discs as previously described. Following the 48 h period, biofilm coated discs were transferred to a new plate and the wells were filled with 180 μL of the solution containing Fluconazole and the EEFLr. Controls were prepared where these contained only the fungal inoculum. The extract and the standard drug were diluted in RPMI 1640. Six hours after Fluconazole and extract addition, the biofilms were quantified using the tetrazolium salt reduction assay, whereby 20 μL, in the 5 μL to 1mL of PBS buffer ratio, sterilized by membrane filtration (0.22 μL pores, Millipore, St. Louis, MO 63103, USA) was added to each well of the microtiter plate, including control wells. The plates were incubated in the absence of light at 37 °C for 18 h. Thereafter, the dye was aspirated and 200 μL of isopropanol was added. The plates were allowed to rest for 15 min then 100 μL of the contents from each well were transferred to a new microtiter plate for reading in a microplate reader with a 570 nm wavelength.

### 4.11. Statistical Analysis

The assays were performed in triplicates for each strain. Statistical analysis was carried out using the GraphPad Prism 6 software (GraphPad Software, Inc., La Jolla, CA, USA). Data are expressed as geometric means. Statistical significance was assessed using a two-way ANOVA (where *p* < 0.05 and *p* < 0.0001 were considered significant and *p* > 0.05 not significant). For the analysis, Tukey’s multiple comparisons test was performed for all means obtained at 5% significance level.

## 5. Conclusions

The *Licania rigida* leaf ethanolic extract analysis by the UPLC-QTOF technique showed the presence of phenolic compounds, previously identified in this genus by other techniques. The EEFLr demonstrated significant antifungal activity with most tested isolates. The extract obtained significant biofilm treatment results for some of the analyzed strains compared to the control drug, Fluconazole, which also obtained relevant results with some of the isolates.

## Figures and Tables

**Figure 1 antibiotics-08-00250-f001:**
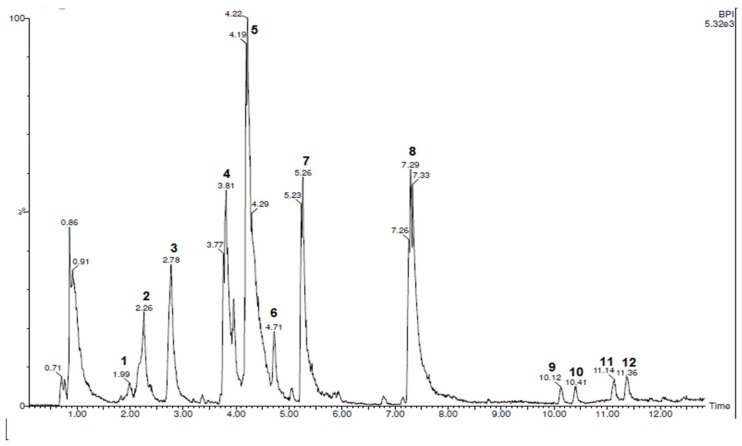
Ultra-performance liquid chromatography with high-resolution mass spectrometry (UPLC—MS) of the EEFLr in negative ionic mode. Peak 1 corresponds to the compounds (epi) gallocatechin Dimer, peak 2 epigalocatechin, peak 3 Gallocatechin, peak 4 Myricetin-O-hexoside, peak 5 Myricitrin and peak 6 Quercetin-O-rhamnoside. Identification of the other peaks is not possible.

**Figure 2 antibiotics-08-00250-f002:**
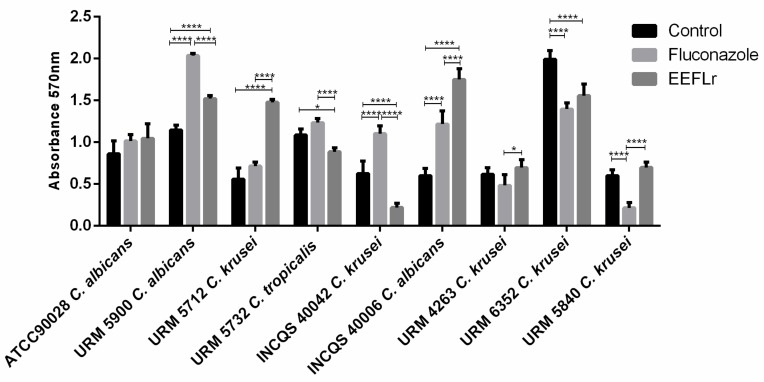
Oxidative activity of biofilm treatment with Fluconazole (64 µg/mL) and Ethanolic Extract of *Licania rigid* Leaves (EEFLr) (2048 µg/mL) formed by clinical *Candida* yeasts. Data represent the mean and standard deviation of absorbance during biofilm production and treatment, compared with the control (biofilm formation without treatment). For the analysis, Tukey’s multiple comparisons test was performed for all means obtained at 5% significance level. The symbols “*” and “****” indicate significant differences between treatments and the control with (*p* ≤ 0.05) and (*p* ≤ 0.0001), respectively.

**Table 1 antibiotics-08-00250-t001:** Analysis of UPLC-ESI-QTOF-MS- Identification of Chemical Constituents of the *Licania rigida* Leaf Ethanol Extract.

PeakNo.	Rt min	[M-H] Observed	[M-H] Calculated	Product Ions (MS/MS)	Empirical Formula	ppm (Error)	Putative Name	References
1	1.99	609.1260	609.1244	305.0717, 441.0822, 423.0754	C_15_H_14_O_7_	2.6	(epi) gallocatechin Dimer (Epigalocatechin)	[14]
2	2.26	305.0655	305.0661	137.0216, 167.0338, 179.0385	C_15_H_14_O_7_	2.0	(L)-Epigalocatechin	[14]
3	2.78	305.0602	305.0661	137.0216, 167.0338, 179.0385	C_15_H_14_O_7_	19.3	Gallocatechin	[14]
4	3.81	479.0828	479.0826	271.0225, 316.0199	C_21_H_19_O_13_	0.4	Myricetin-*O*-hexoside	[15]
5	4.22	463.0879	463.0877	316.0143	C_21_H_20_O_12_	0.4	Myricitrin	[15]
6	4.71	447.0913	447.0927	255.0283, 271.0165	C_21_H_20_O_11_	3.1	Quercetin-*O*-rhamnoside	[15]
7	5.26	351.0147	351.0141	151.0035, 203.9725, 271.0602	C_18_ H_7_ O_8_	1.7	Not identified	-
8	7.29	363.0135	363.0141	267.0294, 268.0337, 283.0612, 347.9891	C_19_H_7_O_8_	1.7	Not identified	-
9	10.12	397.1346	397.1346	125.0241, 183.0080, 277.2179, 311.1738	C_15_H_25_O_12_	3.5	Not identified	-
10	10.41	397.1363	397.1346	183.0066, 235.0800, 277.2123, 325.1740	C_15_H_25_O_12_	4.3	Not identified	-
11	11.14	277.2164	277.2168	183.0114, 184.0016, 253.1086	C_18_H_29_O_2_	1.4	Not identified	-
12	11.36	277.2170	277.2168	183.0114, 184.0264, 253.1031	C_18_H_29_O_2_	0.7	Not identified	-

Rt = retention time; M-H = Mass-to-charge ratio; ppm = parts-per-million.

**Table 2 antibiotics-08-00250-t002:** Minimum Inhibitory Concentration (MIC) (μg/mL) of *Licania rigida* Leaf Ethanol Extract (EEFLr) and Antifungal Fluconazole against the isolates used in this study.

Strains	EEFLr	Fluconazole
*C. albicans* URM5900	≥ 1024 µg/mL	16 µg/mL
*C. krusei* URM5712	256 µg/mL	64 µg/mL
*C. tropicalis* URM5732	≥ 1024 µg/mL	64 µg/mL
*C. albicans* ATCC90028	256 µg/mL	1 µg/mL
*C. krusei* INCQS40042	≥ 1024 µg/mL	64 µg/mL
*C. albicans* INCQS40006	≥ 1024 µg/mL	64 µg/mL
*C. krusei* URM4263	32 µg/mL	64 µg/mL
*C. krusei* URM6352	64 µg/mL	64 µg/mL
*C. krusei* URM584O	32 µg/mL	32 µg/mL

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
