# Peer review of "Evaluation of the Antifungal Activity of the *Licania Rigida* Leaf Ethanolic Extract against Biofilms Formed by *Candida* Sp. Isolates in Acrylic Resin Discs"

_antibiotics, 2019, doi:10.3390/antibiotics8040250_

Round 1

Reviewer 1 Report

The manuscript of Freitas et al. performs the characterization of compounds of the Licania rigida Leaf ethanolic extract by UPLC-QTOF-MS/MS.  In addition the effect of these extracts on the MIC values of several Candida isolates was observed and on the ability to form biofilms. The extract was shown to have some antifungal activity against some of the Candida isolates tested (5 of 9) and to reduce also the biofilm formation of at least 3 isolates, C. tropicalis URM5732 and C. krusei INCQS40042 and URM6352.

Major comments:

The abstract should be re-written. The type of compounds found in EEFLr should be enunciated here. Some results summarized here don’t reflect the results described on the figures presented. Example: “The extract reduced C. albicans biofilm formation better than treatment with fluconazole”. However, none reduced the ability to form biofilms compared with the control, instead exist an increase on biofilm formation with the addition of fluconazole or extract. The same affirmation is done on the results section (Page 4, Line 16-17) and should be corrected. Other example: The last phrase should be more contained, because the extracts only had a significant antifungal activity against 5 isolates and only 3 isolates shown reduced ability to form biofilms after treatment with the extracts. Several empirical formulas shown on Table 1 don’t match with the putative name given (Example peak 1, 5 and 6). Please confirm using PubChem and literature. Re-write the text accordingly in the results section. Table 1: Peak 9 to 12 appears as not identified, but in PubChem exist a match to the empirical formula and to the molecular mass observed. Table 1: The RT min shown on the Table 1 don’t match to the presented on Fig. 1 for Peak 1 to 4, and 9. How the authors explain the increase biofilm formation after the treatment with fluconazole or extract for some Candida Strains? This was previously observed in other studies? Exist any correlation between the MIC values of the extracts and the ability to form biofilms after treatment with the extract? Discuss please. Figure 2 and Figure 3 can be joined in one, because the data presented on figure 2 is the results of the control of Figure 3. In the first paragraph of the Discussion section it should be present the compounds identified for the EEFLr in this work. The first phrase of the second paragraph (Page 6, line 48-49) is speculative, please rephrase. Page 6, line 65-66: The reduction on biofilm formation after fluconazole treatment was only observed for 3 strains. Therefore, please rephrase this sentence. Page 6, Line 70-71: This observation is not correct, because in the present study was not performed combined therapy with fluconazole and extract. Therefore, this interpretation is impossible. Page 6, line 76-78: Not clear the observation, please rephrase. Page 6, line 80-81: This phrase is vague, please add more information. Page 6, line 82-84: Species or Strains? Because different strains of the same species can have different susceptibilities. The authors should compare with other authors that tested the same strains.

Minor comments:

Page2, Line 60: Is missing a reference (Example: Soares Santos et al. 2019). Page 2, Line 67: Meaning of the abbreviation EEFLr (it only appears in Materials and Methods section) Table 1: Please write the data on “Putative name column” in English. Figure 1: The “Peak designation” for each peak on the figure legend will facilitate the interpretation of the Figure by the reader. Table 2: Had to the column “Strains” the Candida Species information and order the strains by species. The table is very confusing without this information. Page 6, line 42: Ref 20 and 21 in the reference list are 22 and 23. Page 6, line 56-57: The Flavonoids action described in this phrase and the reference used is about bacteria or yeasts? Section 4.9 and 4.10: It is not required to mention several manuscripts that used the same method for quantification of biofilms. It should be only referenced the original article that established this protocol and what are the adaptations performed by the authors. Several reference are from manuscripts written only in Portuguese. Please add or substitute for other journals written in English.

Author Response

ANSWERS TO THE REVIEWERS 1

The manuscript of Freitas et al. performs the characterization of compounds of the Licania rigida Leaf ethanolic extract by UPLC-QTOF-MS/MS.  In addition the effect of these extracts on the MIC values of several Candida isolates was observed and on the ability to form biofilms. The extract was shown to have some antifungal activity against some of the Candida isolates tested (5 of 9) and to reduce also the biofilm formation of at least 3 isolates, C. tropicalis URM5732 and C. krusei INCQS40042 and URM6352.

Major comments:

The abstract should be re-written. DONE

The type of compounds found in EEFLr should be enunciated here. DONE

Some results summarized here don’t reflect the results described on the figures presented. Example: “The extract reduced C. albicans biofilm formation better than treatment with fluconazole”. However, none reduced the ability to form biofilms compared with the control, instead exist an increase on biofilm formation with the addition of fluconazole or extract. CORRECTED

The same affirmation is done on the results section (Page 4, Line 16-17) and should be corrected. DONE

Other example: The last phrase should be more contained, because the extracts only had a significant antifungal activity against 5 isolates and only 3 isolates shown reduced ability to form biofilms after treatment with the extracts. DONE

Several empirical formulas shown on Table 1 don’t match with the putative name given (Example peak 1, 5 and 6). Please confirm using PubChem and literature. Re-write the text accordingly in the results section. Table 1: Peak 9 to 12 appears as not identified, but in PubChem exist a match to the empirical formula and to the molecular mass observed. DEAR REVIEWER, ABOUT THE FORMULAE FROM THE PUBCHEM TO THE PEAKS 9 AND 12, THERRE IS NOT HOW TO CONFIRM DUE THE FACT THAT THE uplc TECHNIQUE DO NOT PERMIT IDENTIFY ANY COMPOUND ONLY BY THE EMPIRICAL FORMULAE AND BY THE MOLECULAR MASS. THIS IDENTTIFICATION NEED ALSO THE RETENTION TIME AND THE PATTERN OF FRAGMENTATION. IN TEH PUBCHEM, THERAR ARE NOT DEMONSTRATED THESE FRAGMENTS, BEIND IMPOSSIBLE THE REAL IDENTIFICATION. I HOPE YOUR COMPREHENSION ABOUT THIS TECHNIQUE LIMITATION

Table 1: The RT min shown on the Table 1 don’t match to the presented on Fig. 1 for Peak 1 to 4, and 9. How the authors explain the increase biofilm formation after the treatment with fluconazole or extract for some Candida Strains? This was previously observed in other studies? Exist any correlation between the MIC values of the extracts and the ability to form biofilms after treatment with the extract? Discuss please. POSSIBLY, AFTER THE EXPOSITION TO THE EXTRACT AND TO THE FLUCONAZOLE, THE BIOFILM CELLS WERE CAPABLE TO PRODUCE FILAMENTS AS A TENTATIVE TO DEFENSE AND TO ENHANCE THEIR VVIRULENCE TO OVERLAP TE STRESS AND THE POSSIBLE DEATH BY THE DRUGS. THIS FACT COULD CAUSE NA ENHANCEMENT TO THE BIOMASS AND TO THE METABOLISM OF THIS BIOLFILM. NONE OTHE STUDY IN THE LITERATURE HAS BEEN FOUND WITH THIS KIND OF BIOACTIVITY

Figure 2 and Figure 3 can be joined in one, because the data presented on figure 2 is the results of the control of Figure 3. DONE

In the first paragraph of the Discussion section it should be present the compounds identified for the EEFLr in this work. DONE

The first phrase of the second paragraph (Page 6, line 48-49) is speculative, please rephrase. DONE

Page 6, line 65-66: The reduction on biofilm formation after fluconazole treatment was only observed for 3 strains. Therefore, please rephrase this sentence. DONE

Page 6, Line 70-71: This observation is not correct, because in the present study was not performed combined therapy with fluconazole and extract. Therefore, this interpretation is impossible. CORRECTED

Page 6, line 76-78: Not clear the observation, please rephrase. DONE

Page 6, line 80-81: This phrase is vague, please add more information. DONE

Page 6, line 82-84: Species or Strains? Because different strains of the same species can have different susceptibilities. The authors should compare with other authors that tested the same strains. CORRECTED

 Minor comments:

Page2, Line 60: Is missing a reference (Example: Soares Santos et al. 2019). CORRECTED

Page 2, Line 67: Meaning of the abbreviation EEFLr (it only appears in Materials and Methods section) DONE

Table 1: Please write the data on “Putative name column” in English. DONE

Figure 1: The “Peak designation” for each peak on the figure legend will facilitate the interpretation of the Figure by the reader. DONE

 Table 2: Had to the column “Strains” the Candida Species information and order the strains by species. The table is very confusing without this information. CORRECTED

Page 6, line 42: Ref 20 and 21 in the reference list are 22 and 23. CORRECTED

Page 6, line 56-57: The Flavonoids action described in this phrase and the reference used is about bacteria or yeasts? DONE

Section 4.9 and 4.10: It is not required to mention several manuscripts that used the same method for quantification of biofilms. It should be only referenced the original article that established this protocol and what are the adaptations performed by the authors. Several reference are from manuscripts written only in Portuguese. Please add or substitute for other journals written in English. CORRECTED

Reviewer 2 Report

Authors describe anti fungal activity of the Licania rigida leaf ethanol extract against biofilms formed by Candida isolates. Authors also provide a partial characterization of obtained extract. It is opinion of the reviewer that manuscript is acceptable for publication only upon taking into account some of the minor concerns raised:

First of all, authors determine MIC values of the extract on fungi strains, but they do not comment high values obtained on most of the strains for the extract with respect to those obtained for Fluconazole control compound. This aspect should be extensively  commented in the Discussion section; About anti-biofilm assays performed by MTT, authors are very cryptic in their comments, but it appears like they have contrasting results. In some cases, authors consider as positive a decrease in oxidative activity of fungi cells; in other cases, instead, they consider as positive an increase in oxidative activity. By this way, it is not clear on which strains authors detected significant anti-biofilm activity. This should be more clearly explained in a revised version of the manuscript.

Author Response

ANSWERS TO THE REVIEWERS 2

Authors describe anti fungal activity of the Licania rigida leaf ethanol extract against biofilms formed by Candida isolates. Authors also provide a partial characterization of obtained extract. It is opinion of the reviewer that manuscript is acceptable for publication only upon taking into account some of the minor concerns raised:

First of all, authors determine MIC values of the extract on fungi strains, but they do not comment high values obtained on most of the strains for the extract with respect to those obtained for Fluconazole control compound. This aspect should be extensively  commented in the Discussion section. DEAR REVIEWER, WE ASSAYED IN THIS STUDY 9 DIFFERENT STRAINS. AMONG THEN, 5 STTRAINS PRESENTED A MIC RANGING BETWEEN 256 A 32µG/ML. THE OTHER 4 STRAINS DEMONSTRATED A MIC  ≥ 1024 µG/ML, THAT COULD BE CONSIDERED AS CLIICALLY IRRELEVANTE. WITH THE FLUCONAZOLE, 6 STRAINS WERE RESISTENTE, WITH MICS HIGHER THAN 64 µG/ML. DUE THIS DIFFERENCE, THE FLUCONAZOLE WERE ASSAYED USING A LOWER CONCENTRATION, BASED IN THE CUT POINT OF 64 µG/M LAND THE EXTRACT WAS ASSAYED IN A HIGHER CONCENTRATION DUE THE SAME REASON,BU WITH A DIFERENTE CUT POINT. THIS WAS BASED IN PREVIOUS WORKDS PUBLISHED. BY THIS FACT, IS IMPOSSIBLE TO DISCUSSS THE DIFFERENCE OF THE CONCENTRATIONS USED TO TE EXTRACT AND THE FLUCONAZOLE. BY THE FACTS INFORMED PREVIOUSLY AND DEMONSTRATED IN THE MANUSCRIPT, THE FLUCONAZOLE IS NOT THE BETTER THERAPEUTIC CHOICE AGAINST THESE STRAINS.

About anti-biofilm assays performed by MTT, authors are very cryptic in their comments, but it appears like they have contrasting results. In some cases, authors consider as positive a decrease in oxidative activity of fungi cells; in other cases, instead, they consider as positive an increase in oxidative activity. DEAR REVIEWER, THE RESULTS DEMONSTRATED BY THE mtt ASSAY WWERE DEMONSTRATED IN THE FIGURE 2 AND THE MENTIONS ABOUT THE OXIDATIVE EFFECT WERE REVISED AND CORRECTED.

By this way, it is not clear on which strains authors detected significant anti-biofilm activity. This should be more clearly explained in a revised version of the manuscript. CORRECTED

Round 2

Reviewer 1 Report

The authors addressed some of the reviewer’s comments that in my opinion improved the manuscript. However, some comments were not addressed at all that in my opinion should be taken in consideration and some minor corrections in the new manuscript version are also suggested:

Major comments: Abstract, Page 1, line 34-36: The albicans URM5900 have an increase of biofilm formation with both treatments compared with the control (no treatment). However, according to Fig. 2, C. krusei URM 6352 (and not C. albicans URM5900) have reduced biofilm formation with extract treatment compared with the control. So, the strain here should be C. krusei URM 6352 and not C. albicans URM5900.   Results, Page 4, line 18-19: This observation is not correct. Due to the fact that treatment with extract resulted in more biofilm formation than the control (Fig. 2).   Results, Page 5, line 34-37: This observation is not correct. The treatments with fluconazole don’t result in biofilm reduction (compared with the control) in the case of krusei URM5712 and C. albicans INCQS40006. So these 2 strains should be removed from this phrase. The increase in biofilm formation in these 2 cases was lower than with the treatment with extract (it’s a different observation). And no possible explanation for increase in biofilm formation with fluconazole or extract treatment is discussed on the manuscript. The authors should review the literature to understand if this was previously observed and discuss on the manuscript.   Discussion, Page 6, line 71-73: Again albicans URM5900 have an increase of biofilm formation with the extract. So this observation here is not necessary. The extract worked well against biofilm formation of some C. tropicalis and C. krusei strains. And these results should be enlightened on the discussion section.   Again no discussion about MIC values and Biofilm reduction formation possible correlation was done.

Minor corrections: Abstract, Page 1, line 31-34: Please rephrase. The sentence is too long and confusing.   Abstract, Page 1, line 37-39: In my opinion, this phrase repeats previous results already described in the abstract. So can be deleted.   Figure 2: What is the Control should be explained in the legend. The treatments are explained on the legend, but not the control.   Page 6, line 81:”Strain”, means one strain in particular, the two strains before enunciated or the krusei species. Please clarify on the text.

Author Response

ANSWERS TO THE REVIEWERS 1

The authors addressed some of the reviewer’s comments that in my opinion improved the manuscript. However, some comments were not addressed at all that in my opinion should be taken in consideration and some minor corrections in the new manuscript version are also suggested:

Major comments:

Abstract, Page 1, line 34-36: The albicans URM5900 have an increase of biofilm formation with both treatments compared with the control (no treatment). However, according to Fig. 2, C. krusei URM 6352 (and not C. albicans URM5900) have reduced biofilm formation with extract treatment compared with the control. So, the strain here should be C. krusei URM 6352 and not C. albicans URM5900. CORRECTED

Results, Page 4, line 18-19: This observation is not correct. Due to the fact that treatment with extract resulted in more biofilm formation than the control (Fig. 2).  CORRECTED

 Results, Page 5, line 34-37: This observation is not correct. The treatments with fluconazole don’t result in biofilm reduction (compared with the control) in the case of kruseiURM5712 and C. albicans INCQS40006. So these 2 strains should be removed from this phrase. The increase in biofilm formation in these 2 cases was lower than with the treatment with extract (it’s a different observation). CORRECTED

 And no possible explanation for increase in biofilm formation with fluconazole or extract treatment is discussed on the manuscript. The authors should review the literature to understand if this was previously observed and discuss on the manuscript.   CORRECTED

Discussion, Page 6, line 71-73: Again albicans URM5900 have an increase of biofilm formation with the extract. So this observation here is not necessary. THIS AFFIRMATION WAS DELETED

The extract worked well against biofilm formation of some C. tropicalis and C. krusei strains. And these results should be enlightened on the discussion section.   CORRECTED

Again no discussion about MIC values and Biofilm reduction formation possible correlation was done. CORRECTED

Minor corrections: Abstract, Page 1, line 31-34: Please rephrase. The sentence is too long and confusing.   CORRECTED

Abstract, Page 1, line 37-39: In my opinion, this phrase repeats previous results already described in the abstract. So can be deleted.   DELETED

Figure 2: What is the Control should be explained in the legend. The treatments are explained on the legend, but not the control.   CORRECTED

Page 6, line 81:”Strain”, means one strain in particular, the two strains before enunciated or the kruseispecies. Please clarify on the text. CORRECTED

Round 3

Reviewer 1 Report

The authors addressed the reviewer’s comments, improving the manuscript and in my opinion aceptable for publication.  The following minor corrections in the new manuscript version are suggested:

Discusssion, Page 6, line 71: Please change "species" to "isolates".  Discussion, Page 6, line 76-77: Please rephrase.

Author Response

ANSWERS TO THE REVIEWERS 1

The authors addressed the reviewer’s comments, improving the manuscript and in my opinion aceptable for publication.  The following minor corrections in the new manuscript version are suggested:

Discusssion, Page 6, line 71: Please change "species" to "isolates".  CORRECTED

Discussion, Page 6, line 76-77: Please rephrase. CORRECTED